# Re-Emerged Genotype IV of Japanese Encephalitis Virus Is the Youngest Virus in Evolution

**DOI:** 10.3390/v15030626

**Published:** 2023-02-24

**Authors:** Guanlun Xu, Tingting Gao, Zhijie Wang, Jun Zhang, Baoqiu Cui, Xinxin Shen, Anyang Zhou, Yuan Zhang, Jie Zhao, Hong Liu, Guangdong Liang

**Affiliations:** 1Shandong Provincial Research Center for Bioinformatic Engineering and Technique, School of Life Sciences, Shandong University of Technology, Zibo 255049, China; 2State Key Laboratory of Infectious Disease Prevention and Control, National Institute for Viral Disease Control and Prevention, Chinese Center for Disease Control and Prevention, Beijing 102206, China

**Keywords:** Japanese encephalitis virus, genotype IV, emerging, evolution

## Abstract

An outbreak of viral encephalitis caused by a Japanese encephalitis virus (JEV) genotype IV infection occurred in Australia between 2021 and 2022. A total of 47 cases and seven deaths were reported as of November 2022. This is the first outbreak of human viral encephalitis caused by JEV GIV since it was first isolated in Indonesia in the late 1970s. Here, a comprehensive phylogenetic analysis based on the whole genome sequences of JEVs revealed it emerged 1037 years ago (95% HPD: 463 to 2100 years). The evolutionary order of JEV genotypes is as follows: GV, GIII, GII, GI, and GIV. The JEV GIV emerged 122 years ago (95% HPD: 57–233) and is the youngest viral lineage. The mean substitution rate of the JEV GIV lineage was 1.145 × 10^−3^ (95% HPD values, 9.55 × 10^−4^, 1.35 × 10^−3^), belonging to rapidly evolving viruses. A series of amino acid mutations with the changes of physico-chemical properties located in the functional important domains within the core and E proteins distinguished emerging GIV isolates from old ones. These results demonstrate the JEV GIV is the youngest JEV genotype at a rapid evolution stage and has good host/vector adaptability for introduction to non-endemic areas. Thus, surveillance of JEVs is highly recommended.

## 1. Introduction

Japanese encephalitis (JE) is an acute neurological infectious disease caused by the Japanese encephalitis virus (JEV). JE is transmitted through arthropod bites and has a mortality rate of about 30%. Furthermore, about 45% of JE survivors have different degrees of central nervous system sequelae, indicating that JE is a public health concern [1]. JEV infects about 68,000 people and causes about 13,600–20,400 deaths yearly, based on the latest data from the World Health Organization (WHO) [2]. About 3 billion people live in JEV epidemic areas. JEV originated from Southeast Asia and is mainly prevalent in Asia. Nonetheless, JEV has been detected in the South Pacific region of Australia and on the west coast of Africa.

JEV is a single-stranded positive-sense RNA virus belonging to the Flaviviridae subfamily, Flavivirus genus. The genome length of JEV is about 10,965 nt and contains a single open reading frame (ORF) encoding a polyprotein that is processed into three structural (C, M, and E) and seven nonstructural (NS1, NS2A, NS2B, NS3, NS4A, NS4B, and NS5) proteins [3,4]. The 5′ and 3’ untranslated regions (UTR) are located on both sides of the ORF and contain abundant cis-acting elements that are crucial for virus replication, immunity, and pathogenicity [5]. Previous studies have indicated three domains in E protein, and the E protein domain III, which has the main cites for neutralizing antibodies, is considered to play a key role in activating receptor and membrane fusion [6,7].

JEVs originated from the Malaysia/Indonesia region [8], then spread to the Indo-China Peninsula, from there, the JEVs were further transmitted to the eastern, central, and western regions of the Asian mainland [9]. JEVs could be divided into five genotypes (GI~GV) with different evolution characteristics, geographical distribution, and host vectors. GI and GIII were once the main epidemic genotypes of the JEV and are widely distributed in Asia and Northern Australia. The hosts of GI and GIII include mosquitoes, pigs, midges, ticks, bats, and humans [10]. GI has gradually replaced JEV GIII in the Asian mainland since 1990s, and has become the dominant genotype [11]. The phylogenetic analysis has shown that GI is the youngest genotype of the JEV, with diversified genetic diversity and advantages in natural selection. GII was once isolated from mosquitoes in Southeast Asia and northern Australia, from 1951 to 1999 [12,13]. GV is the oldest lineage in the JEV populations, and was initially isolated from a patient in Malaysia in 1952 [14]. Then, JEV GV was not reported again until about 60 years later, when it was isolated from mosquitoes in Tibet, China in 2009 [15,16] and in South Korea in 2011 [17,18]. JE outbreak was reported in Korea in 2015 [19], and about 20 strains of JEV GV have been reported in mosquitoes and in a clinical case between 2010 and 2021 in Korea [17,18,19,20,21]. The JEV has also been detected in southward regions. Three cases of viral encephalitis were found on Badu Island in the Torres Strait, Northern Australia, between February and March 1995, all of which were confirmed to be infected with the JEV and two of them died. Since then, ten strains of the JEV have been isolated from mosquitoes and patients in this region, representing the first case of JE in Australia [12]. A total of 44 JEV isolates were isolated from Culex tritaeniorhynchus and one isolate was isolated from pigs in Australia in 1998 [22]. Molecular genetic studies showed that all JEV isolates from JE patients, pigs, and mosquitoes in 1995 and 1998 in Australia belonged to JEV GII.

JEV GIV was a long-neglected genotype and was initially isolated from mosquitoes in Indonesia in the 1980s [8]. GIV was not detected in natural hosts and vectors worldwide for about half a century, indicating that it might have become extinct. However, GIV appears to be active in nature again, several JEV GIV isolates have been reported in Indonesia starting in 2020 [23,24,25]. Even an outbreak of the JEV caused by JEV GIV occurred in Australia between 2021 and 2022, suggesting that JEV genotype IV is re-emerging [26,27]. Notably, the origin and evolutionary characteristics of JEV GIV, and the genetic relationship between the re-emerging and the original JEV GIV strain is unclear. This study aimed to systematically analyze the molecular evolution of the JEV GIV strains.

## 2. Materials and Methods

### 2.1. JEV Analysis Pipeline Design

First, the JEV whole-genome sequence dataset was constructed using 67 strains sequenced in our lab and 48 strains obtained from the NCBI database. Sequence inclusion criteria are described in the dataset construction section. The phylogenetic and evolutionary dynamics were analyzed based on the whole JEV genome dataset. The phylogenetic and evolutionary dynamics of JEVs were then evaluated, especially for GIV JEVs. Detailed analysis was performed to see whether GIV JEVs had a genetic diversity to identify the differences between different lineages of JEV GIV. The whole genome dataset was split into 12 parts based on the genome structure of 5’UTR, ORF (C, M, E, NS1, NS2A, NS2B, NS3, NS4A, NS4B, and NS5), and 3’UTR gene datasets for the general sequence feature analysis and higher-level structure analysis. The primary sequence analysis entailed sequence similarity, nucleotide, and amino acid mutations. Finally, vaccine protective ability was evaluated via the envelope protein structure comparison between vaccine strains and emerging GIV isolates. The software used in the analyses is shown in the flow chart of the JEV phylogeny and genome sequence feature analysis pipeline (Appendix A). The datasets, sequence alignments, and the software used in this study are available from the corresponding authors upon request.

### 2.2. Dataset Construction in the Analyses

The whole genome sequences of the JEV were downloaded from the GenBank database (as of November 2022) to understand the evolutionary relationship among the JEV strains, the origin of the populations in nature, and the latest characterized evolutionary dynamics. Only sequences with clear and comprehensive background information on the JEV, including isolation location, isolation time, and host (vector), were included. Sequencing data of unhelpful, derivative, and vaccination strains were removed from the dataset. Sequence dataset was compared using T-COFFEE software [28]. Strains with a sequence similarity greater than 98% were excluded from the dataset.

### 2.3. Time Scaled Phylogenetic Analysis of the JEV

A total of 115 whole genome sequence datasets of JEV strains obtained from the NCBI database and sequenced in our laboratory were screened for the phylogenetic analysis to assess the evolutionary characteristics of the origin of the JEV populations in nature. The JEV sequence recombination signal analysis was performed using RDP3 (Recombination Detection Program3) [29]. Phylogenetic signal detection was performed using IQ-tree [30] in Phylosuite software [31], and Xia’s test was applied for the nucleotide substitution saturation detection [32] to determine the suitability of the dataset for phylogenetic analysis. ModelFinder [33] in Phylosuite software was used to detect the screening of its optimal base substitution model, and the BEAST software package [34] was used to carry out the evolutionary dynamics and evolutionary rate analysis of the JEV populations. The parameters were selected from a combination of relaxed (uncorrelated log-normal) molecular clock model and different population evolution models for the JEV evolutionary rate and evolutionary time analysis, with the chain length set to 2 × 10^9^ to ensure effective mixing. The best matching model group was determined by the Bayes factor (BF) test. The convergence of the parameters was tested with Tracer software [35], and the results were expressed as the effective sample size (ESS) > 200. TreeAnnotator software was set to annotate the maximum clade credibility (MCC) tree after 10% sampling aging, and Figtree software (http://tree.bio.ed.ac.uk/software/figtree/, accessed on 5 January 2023) for visualization of the MCC tree. Furthermore, the population’s dynamics were inferred by the Bayesian skyline reconstruction.

### 2.4. Analysis of the Geographic Migration of JEV GIV

The Bayesian random search variable selection (BSSVS) analysis was performed using the BEAST package to elucidate the geographic migration history of GIV JEVs. The method can estimate the most likely state of each node in the MCC tree, allowing to reconstruct the ancestral position of the ancestral virus lineage along the phylogenetic tree. ModelFinder in the Phylosuite software was used to calculate the Tn93 + F + G4 alternative model as the optimal model. Each region in the phylogeographic reconstruction is coded as a discrete trait. The uncorrelated relaxed clock (URC) model and constant size (CS) tree prior model were chosen, and the chain length was set to 500 million. A tracer software was used to analyze the convergence of the parameters, and the results were expressed as ESS > 200. The MCC tree was annotated using TreeAnnotator software after 10% of burn-in. SPREAD3 software was used to output BSSVS of continuous diffusion processes as HTML files [36]. Firefox was used to visualize the geographic migration history.

### 2.5. Phylogenetic Analysis Based on the Variable Region within 3’UTRs of JEVs

Our previous study [5] identified that a 300 nt-length region located at the beginning of the JEV 3’UTR, termed variable region (VR), exhibited significant genotypic specificity. In this study, the VR region of the 115 JEV sequences was also analyzed to verify whether the genomic sequences of the emerging JEV GIV conform to the phylogenetic characteristics of the JEV population. The BEAST software package was used to conduct the phylogenetic analysis, a BF test was used to determine the best matching combination of the molecular clock and tree prior models. The uncorrelated relaxed clock model with the constant size tree priori model were selected, with the chain length of 10 million.

### 2.6. Sequence Analysis of GIV JEVs

#### 2.6.1. Analysis of the Whole Genome Sequence Similarity of GIV JEVs

The JEV vaccine strain p3 was used as the standard sequence, while the whole genome sequences of all JEV GIV strains formed a dataset. Simplot software [37] was used for the comparative analysis and visualization of the global sequence similarity. The similarity curve between the sequences of the GIV strains was plotted via the evolutionary distance algorithm (Kimura two-parameter model) with 20 nt in each step and a 200 nt window.

#### 2.6.2. Analysis of the Nucleotide and Amino Acid Similarity in Different JEV Genotypes

Mafft software (https://mafft.cbrc.jp/alignment/software, accessed on 5 January 2023) was used to align the JEV genome sequence. Bioedit software was used for sequence editing. The nucleotide and amino acid similarity analysis of the JEV whole genome sequences were completed using the DNAStar software.

#### 2.6.3. Comparison of the Sequence and Structure of the 5′ and 3′ UTRs of JEVs

Mfold Web server (http://unafold.rna.albany.edu/q=mfold, accessed on 15 January 2023) website and the VARNA program were used to perform and visualize the secondary structure modeling of the UTRs of the representative strains of JEVs. The original JEV GIV isolate (JKT6468), and the emerging GIV isolate (Bali93) were selected for the higher-order structure analysis of UTRs. The JEV standard strain (Nakayama) was used as the reference.

#### 2.6.4. Analysis of Amino Acid Mutations of Each Protein of GIV JEVs

A dataset of the JEV GIV strains was constructed with the JEV vaccine strain p3 strain as the standard. Mafft software (https://mafft.cbrc.jp/alignment/software, accessed on 5 January 2023) was used for the JEV genome sequence alignment. The sequences were edited using Bioedit software. Sequence divergence sites were displayed and analyzed using GeneDoc software.

### 2.7. Comparative Structural Analysis of the Envelope Protein of Emerging GIV JEV Isolates and Vaccine Strain P3

The JEV vaccine strain P3 and representative strains of the emerging JEV GIV isolates from Indonesia (Bali93) and Australia (NT_Tiwi) were selected for a secondary structure, three-dimensional structure, and surface charge density comparison analyses. Homology modeling was performed using YASARA software [38] with PDBID: 5WSN as a template to obtain the protein monomer structures of the JEV isolates p3, Bali93, NT_Tiwi. PyMol and YASARA software were further used for the structures and surface charge density comparison analyses.

## 3. Results

### 3.1. Complete Genome Sequence Dataset Construction

The final dataset contained 115 whole genome sequences of the JEV (Appendix A), including 53 strains of GI, one strain of GII, 48 strains of GIII, seven strains of GIV, and six strains of GV. The strains were mainly detected in mosquitoes (n = 54), humans (n = 34), pigs (n = 24), cattle (n = one), horses (n = one), and midges (n = one) samples. The JEV strains were isolated from 1935 to 2022, and the virus isolation site contained all of the regions where the JEV had been reported (from latitude 15 °S to latitude 45 °N) (Detailed information is shown in Table 1).

### 3.2. Time-Scaled Phylogenetic Evolutionary Analysis

The Bayesian skyline model with a relaxed molecular clock was selected as the best fit model according to the Bayes factor and 95% HPD intervals. Based on the Bayesian Markov chain Monte Carlo (MCMC) analysis, the MCC tree was established (Figure 1). The posterior probability value of each branch node was greater than 0.75, indicating the robustness of the result. The MCC tree showed that the tMRCA (time to most recent common ancestor) of the JEV was estimated to be about 1037 years ago (95% HPD: 463–2100) and then differentiated at least five times. The tMRCA of each genotype appeared in the following order: GV: 245 (95% HPD, 96–536); GIII: 179 (95% HPD, 124–260); GI: 132 (95% HPD, 82–212); and GIV: 122 (95% HPD, 57–233). As only one JEV GII sequence was available, the tMRCA of GII was not estimated. Thus, JEV GIV represents the youngest lineage within the JEV populations.

Further analysis revealed that GIV evolved into two sub lineages: the Old sub lineage consists of two JEV GIV strains isolated around the 1980s (VN 113, 1979; JKT6468, 1981) and the corresponding emerging sub lineage. The emerging sub lineage is further clustered into Indonesian and Australian clades based on the location of isolation. The Indonesian clade consists of three isolates from pig specimens (JEV/sw/Bali/93/2017), mosquitoes (19CxBa-83-Cv, 2019), and human (Bali 2019). The Australian clade consists of two isolates from patients in Australia (JEV/Human/NT_Tiwi-Islands/2021) and diseased pigs in Australia (JEV/sw-22-00722-11/Qld/2022).

The analysis showed that the tMRCA of the old and emerging GIV sub lineages were 54 years ago (95% HPD, 44–71) and 62 years ago (95% HPD, 23–113), respectively. The Indonesian and Australian clades appeared 22 years ago (95% HPD, 10–39) and 6 years ago (95% HPD, 2–14), respectively.

### 3.3. Evolutionary Rate and Population Dynamic Analysis

The mean nucleotide substitution rate for the entire JEV population was estimated at 2.75 × 10^−4^ substitutions per site per year (s/s/y) (95% HPD values, 1.62 × 10^−4^, 3.97 × 10^−4^). For JEV GIV, the estimated rate was 1.145 × 10^−3^ (95% HPD values, 9.55 × 10^−4^, 1.35 × 10^−3^).

The population dynamics diagram (Figure 2) showed that the JEV populations could have undergone relatively complex changes during evolution. The population diversity of the JEV population slightly increased from 1945 to 1965, it significantly increased from 1965 to 1975, then decreased from 1975 to 2000, and then slightly increased from 2000 to 2010, after which it remained relatively stable.

### 3.4. The Dispersal Route of JEV GIV Based on Phylogeographic Analysis

The estimated history of the JEV GIV dispersal route over time was shown in Figure 3. According to the results of the phylogeographic analysis, the JEV GIV was originated in the Vietnam/Indonesia region (95° E–141° E, 23° N–9° S) around the 1980s and was limited to this region for the next 20 years. JEV GIV then spread from the Vietnam/Indonesia region to the Tiwi Islands, Northern Territory, Australia (130° E–131° E, 11° S–12° S) from 2005 to 2020, then further spread to Queensland, Australia (130° E–131° E, 11° S–12° S), from 2020 to 2022.

### 3.5. Phylogenetic Analysis of the Variable Region of JEVs

The phylogenetic tree (Figure 4) based on the VR region could be divided into five genotypes. The topology of the phylogenetic trees based on the VR region and whole genome sequences of the 115 JEV strains were identical.

### 3.6. Similarity Analysis of the Nucleotides and Amino Acids of Different JEV Genotypes

The nucleotide and amino acid similarity of GI are 92.5–100% (average: 97.1%) and 96.9–100% (average: 99.2%), respectively. The nucleotide and amino acid similarity of GIII are 94.0–99.9% (average: 97.4%) and 97.1–100% (average: 99.0%), respectively. The nucleotide and amino acid similarity of GIV are 94.3–99.7% (average: 96.4%) and 97.9–99.9% (average: 98.9%), respectively. The nucleotide and amino acid similarity of GV are 90.5–100% (average: 94.3%) and 98.1–100% (average: 98.9%), respectively (Appendix A). There was only one GII strain, which was not compared. The nucleotide similarity was highest in GIII, followed by GI, GIV, and GV. Amino acid similarity was highest in GI, and it was lowest in GIV and GV. The average similarity values of 5’UTR, C, M, E, NS1, NS2A, NS2B, NS3, NS4A, NS4B, NS5, and 3’UTR genes and their encoded proteins of five JEV genotypes are shown in Appendix A.

### 3.7. Sequence Analysis of GIV JEVs

#### 3.7.1. Similarity Analysis of the Whole Genome Sequence of GIV JEVs

The nucleotide similarity of the old and emerging sub lineages of GIV JEVs are 99.1% and 95.9–99.7% (average: 97.35%), respectively. Furthermore, nucleotide similarity of Indonesian and Australian clades is 98.1–99.2% (average: 98.6%) and 99.7%, respectively. The nucleotide similarity between vaccine strain p3 and GIV strains range from 83.8% to 84.9% (average: 84.7%) (Figure 5) (Appendix A).

The nucleotide similarity plot analysis between the JEV GIV and P3 strains are shown in Figure 6. The nucleotide sequences of JEV GIV and P3 are significantly different. Furthermore, the nucleotide similarities of the genes encoding C, E, NS2A, and NS5 proteins are less than 75%. The emerging and old sub lineages of GIV JEVs are different in 5’UTR, C, E, NS3, and NS5 genes.

#### 3.7.2. Comparative Analysis of the Sequence and Structure of the UTRs of JEVs

Three of seven GIV JEVs have completed gene sequence information in 5′UTR, and four isolates have gene deletion at the 5′ end. The complete sequence length of all GIV isolates at the 5’UTR region is 95 nt. As the gene sequence encoding the C protein also participates in the formation of the high-order structure of 5′UTR, thus this sequence segment was also included in the analysis (146 nt).

The primary structure analysis showed that GIV strains and the JEV standard strain (Nakayama) have 14 different nucleotide sites (Figure 7A). Further analysis showed that the JEV GIV old sub lineage representing strain JKT6468 have eight different nucleotide sites when comparing with the other GIV isolates. (G27A, C69A, C77T, A81G, T93A, C111G, T124A, and T138A) (Appendix A). The Australian strains (JEV/Human/NT_Tiwi-Islands/2021, JEV/sw-22-00722-11/Qld/2022) have a site of difference from other strains (T122G) (Figure 7A).

The secondary structure and the cis-acting elements of the GIV emerging sub lineage representing strain (JEV/sw/Bali/93/2017) and the JEV standard strain (Nakayama) were highly similar. All of them had stem-loop structures (SL) consisting of 10 stems (H1–10), six interior loops (I) (I1–6) and four terminal loops (T) (T1–4). It includes SLA and SLB, where SLA is composed of H1-H6, T1-T2, and I3-I6, while SLB is composed of H7-H8, T3 and I2, and H9-H10, and T4 and I1 are composed of the capsid-coding region hairpin element (cHP). However, the secondary structure of the old sub lineage JEV GIV (JKT6468) is significantly different from that of the standard strain, and its secondary structure is composed of only 6 H (H1–6), 3 I (I1–3), and 4 T (T1–4). To form SL, SLA consists of H1-H3, T1-T2, and I2-I3. SLB consists of H4 and T3 only. H5-H6, T4, and I1 constitute cHP.

In 3 ‘UTR (Figure 7B), four JEV GIV strains had complete 3’UTR sequences. The length of the 3′UTR of different strains varied from 579–601 nt. There were 59 nucleotide differences between JEV GIV isolates and the standard JEV strain (Nakayama). The emerging and the old sub lineage JEV GIV have a deletion of seven nucleotides in the variable region at positions 8–15 of JKT6468 and seven common nucleotide mutation sites (G54A, C143U, A238G, U240A, C333U, G414A, and C463U). The Australian and Indonesian strains have five common nucleotide mutation sites (A59G, C251U, U272C, G318A, and C380U) (Figure 7B).

JEV 3’UTR consists of three domains (I, II, and III), within which the cis-acting elements including 3’VVR, xrRNA1, 3’vrSL, xrRNA2, DB1, DB2, 3’sHP, 3’SL, PK, 3’RCS, 3’CS, 3’cCS, 3’DAR, and 3’UAR are located. The secondary structures of the JEV GIV representing strains and the standard JEV strain (Nakayama) are different (Figure 7B). The secondary structures of 3’UTR (Figure 7B) are divergent among JEV GIV isolates and the Nakayama strain. The secondary structural elements of the Nakayama strain have 35 H, 17 T, and 18 I. The old sub lineage GIV (JKT6468) has 38 H, 16 T, and 21 I while the emerging sub lineage GIV (JEV/sw/Bali/93) possesses 33H, 16T, and16I.

Further, the secondary structures of the VR region are significantly different between the emerging and old sub lineage representing strains (JEV/sw/Bali/93/2017 and JKT6468), whereas the VR regions are divergent among the JEV genotypes, but conserved within the same genotype. The 3’UAR, associated with genomic cyclization, is conserved within the JEV genotypes. We found that nucleotide composition and even the location of 3’UAR were divergent between the old and emerging sub lineage GIV isolates.

#### 3.7.3. Analysis of the Mutation Sites of Proteins within GIV JEVs

The genes encoding three structural proteins and seven non-structural proteins were aligned and analyzed using the JEV vaccine strain P3 as a standard. A total of 198 amino acid mutations were detected in GIV JEVs (Appendix A).

There were 21 mutations located in the C protein, of which 19 were completely consistent within the emerging sub lineage of GIV. The old sub lineage isolate (VN_113) has six mutations compared with the emerging sub lineage while JKT6468 has twelve mutations. Notably, there were seven mutations of the emerging sub lineage GIV strains with changed amino acid physicochemical properties when compared with the old sub lineage representing isolate JKT6468. For example, position 6 is changed from basic arginine to neutral glycine, position 10 is changed from neutral hydrophobic isoleucine to basic hydrophilic lysine, position 41 is changed from neutral hydrophobic isoleucine to basic hydrophilic arginine, position 52 is changed from hydrophilic serine to hydrophobic alanine, position 60 is changed from hydrophilic serine to hydrophobic alanine, position 112 is changed from hydrophilic threonine to hydrophobic valine, and position 122 changed from hydrophobic alanine to hydrophilic glycine (Appendix A).

A total of 24 mutations were detected in the PrM protein, of which 23 are completely consistent within the emerging sub lineage. The old sub lineage isolates (VN_113) has four mutations compared with the emerging sub lineage, while JKT6468 has 10 mutations, indicating that VN_ 113 is closer to the new lineage strains in the PrM protein.

There were 35 mutations located in the E protein, of which 26 were completely consistent within the emerging sub lineage. Furthermore, seven of nine mutations were different from the Australian clade and the others, indicating that the Australian clade is divergent. There were 12 mutations with significant amino acid physicochemical property changes in the E protein among the p3 strain and the emerging sub lineage GIV JEV isolates. For T46I, A295T, G306E, A366S, E388G, L408S, V490T, and S327Q have the type of significant hydrophilic and hydrophobic change. The steric hindrance effects of amino acid side chains are different for P227S, P228S, A399P, and G306E.There were significant charge changes in N36H, G306E, and E388G.

There were in total 118 amino acid mutations located in seven non-structural proteins. The detailed information was listed in Appendix A.

### 3.8. Comparative Analysis of the Envelope Protein between Emerging JEV GIV Isolates and Vaccine Strain P3

The JEV E protein consists of three main domains (DI, DII, DIII) and a helical stem and anchor regions (Figure 8). Herein, the secondary structures were annotated based on the crystal structure of the JEV E protein revealed in the distribution of amino acid residues in the three domains. DI, as the central domain, consists of E1-E50, E132-E190, and E284-E290; DII (ductile domain) consists of E53-E130 and E201-E280; and DIII (globular domain) consists of E308-E398. The sequence and secondary structure (Figure 8) of the vaccine strain p3 were aligned with the representative strains of the emerging sub lineage GIV (Indonesian isolate JEV/sw/Bali/93/2017, Australian isolate JEV/Human/NT_Tiwi -Islands/2021), and results identified 26 amino acid residue mutations between P3 and emerging GIV strains. A total of 21 conservative mutations were identified in the alignment of the GIV representative strains and the P3 strains of the JEV (A15V, N36H, T46I, M76T, A129T, P227S, P228S, S230N, G261A, A295T, G306E, S327Q, A351V, A366S, E388G, A399P, L408S, L482M, A486V, V490T, and V492L). Three mutations are located in DI β-sheets (A15V, N36H, and T46I); six mutations (M76T, A129T, P227S, P228S, S230N, and G261A) are located in DII and four mutations in the DIII domain (S327Q, A351V, A366S, and E388G). The DIII region has an immunoglobulin-like fold structure, which is located at the C end of the extracellular domain and connects with DI through a short peptide connector. The mutation sites A295T and G306E are located in the linker region connecting DI and DIII.

The three-dimensional structures and electrostatic potential analysis have also been conducted to further explore whether these mutations affect the higher order structures and surface charge density of the E protein. A total of 12 amino acid mutations with significant property changes had been labeled in the structures of the E protein (Figure 9). There were great differences in the R group of the amino acid and the electrical polarity of the 12 amino acid residues, resulting in significant structural and electronic charge differences at these sites between the P3 and emerging GIV isolates (Figure 9).

## 4. Discussion

This study provides comprehensive and detailed bioinformatics analysis of the whole genome sequences of the JEV strains isolated from patients, animal hosts, and kinds of vectors worldwide over 80 years. Our results lead to an estimate that the tMRCA of JEVs occurred 1037 years ago (95% HPD: 463–2100) and diverged into five genotypes in the sequence GV, GIII, GII, GI, and GIV, with GIV being the youngest and with the fastest evolutionary rate. Although previous studies have estimated the tMRCA of JEVs [11,39] and revealed that GI was the youngest within the JEV population based on the JEV’s whole genome sequence dataset with only one strain of GIV. The results obtained may not be accurate because of the lack of the whole genome sequence information of GIV. In this study, the updated complete genome sequence information of all GIV JEVs was used for analysis. These GIV isolates include the old sub lineage isolates (from the 1980s) and the emerging sub lineage isolates (from 2016 and 2022). It not only complements the lack of limited GIV sequence information, but also the long span of virus isolation time from 1980 to 2022, makes the phylogenetic estimation more robust. Chisha Sikazwe et al. [27] estimated the evolutionary history of JEVs using 519 spatio-temporally-defined complete E gene sequences. The topology of their MCC tree based on the E gene dataset was similar to that of the MCC tree based on the whole genome sequence of JEVs in this study. According to the phylogenetic analysis based on the whole genome or the E protein gene sequences, these results indicate that GIV is the youngest among the five genotypes.

JEV, West Nile virus (WNV), yellow fever virus (YFV), dengue virus (DENV), tick-borne encephalitis virus (TBEV), and Zika virus (ZIKV) are the most prevalent and severe arboviruses worldwide [40,41]. WNV, YFV, TBEV, and ZIKV appeared about 200 years ago [42], 306 years ago [43], 664 years ago [44], and 155 years ago [45], respectively. The most common ancestor of JEV GIV appeared 62 years ago, suggesting it belongs to the youngest viruses in mosquito-borne Flaviviruses.

The nucleotide substitution rate can be used to assess the evolutionary characteristics of viruses, as the high mutation rate often leads to strong adaptability and high pathogenicity [46]. The average nucleotide substitution rates for the WNV, TBEV, and YFV are estimated to be 5.06 × 10^−4^ s/s/y [47], 2.104 × 10^−4^ s/s/y [48], and 4.2 × 10^−4^ s/s/y [43], respectively. The nucleotide substitution rate of the JEV GIV population is about 1.145 × 10^−3^ (95% HPD values, 9.55 × 10^−4^, 1.35 × 10^−3^), making it the currently known fast-evolving virus among mosquito-transmitted Flaviviruses.

Notably, the nucleotide substitution rate of the emerging lineage GIV is equivalent to the that of the Asian lineage of ZIKV, which caused the outbreak of ZIKV infection in 2016 [49] (nucleotide substitution rate: 1.0233 × 10^−3^ s/s/y: 95% HPD, 8.2 × 10^−4^ to 1.2 × 10^−3^). GIV JEV and Asian lineage of ZIKV are both emerging pathogens with a high nucleotide substitution rate. High mutation rates coupled with rapid replication could enhance the viral adaptability and competitiveness, as well as pathogenicity. This might be one of the contributing factors of the JEV GIV infection outbreak in Australia.

GI has gradually replaced JEV GIII in the Asian mainland since the 1990s, and has become the dominant genotype [11]. JEV GV was first isolated from patients in Malaysia in the 1950s [50] and identified as JEV GV in 1994 [8,51]. JEV GV was not reported again until about 60 years later, when it was isolated in mosquitoes in Tibet, China in 2009 [15,16] and in South Korea in 2011 [17,18]. JE outbreak was reported in Korea in 2015 [19]. The JEV has also been detected in southward regions. Three cases of viral encephalitis were found, all of which were confirmed to be infected with the JEV and two of them died, on Badu Island in the Torres Strait, Northern Australia, between February and March 1995. Since then, ten strains of the JEV have been isolated from mosquitoes and patients in this region, representing the first case of JE in Australia [12]. A total of 44 JEV isolates were isolated from *Culex tritaeniorhynchus* and one isolate was isolated from pigs in Australia in 1998 [22]. Molecular genetic studies showed that all JEV isolates from JE patients, pigs, and mosquitoes in 1995 and 1998 in Australia belonged to JEV GII. From 2000 to 2002, JEVs were still active in Australia, a total of 43 isolates were isolated from *Cx. Annulirostris* and one was from *Ochlerotatus vigilax*. Three strains of the JEV were isolated from sentinel pig samples, and one from *Cx. Gelidus* but no case of JE was found in the local populations [22]. A strain of JEV GI was isolated from pigs in Australia in 2000 [22], indicating there were two JEV genotypes circulating in Australia, and the JEV may be transmitted through monsoons to Australia [52]. As previously mentioned, the outbreak of JE occurred in North-Eastern Australia in 2021–2022 and was confirmed to be caused by JEV GIV. This was not only the first discovery of this JEV genotype in Australia, but also the first outbreak of human viral encephalitis caused by JEV GIV [26,27].

A previous study [8] demonstrated that the ancient lineages of JEV GIV (JKT6468) possess an unusual series of amino acids at the carboxyl terminus of the core protein together with a signature amino acid mutation (E327) in the envelope protein, altering the vector and/or host preference, which could have contributed to the limited host and vector ranges and remain geographically confined to the Indonesia-Malaysia region [8].

Interestingly, JEV GIV was isolated successively in Indonesia and Australia in the South Pacific after 2017, and JEV GIV even caused JE outbreak in Australia in 2022. We examined the old and emerging sub lineages of GIV JEVs’ complete genome sequence to look for molecular determinants that might relate to the differences in geographical distribution and host preference. In this study, our findings showed that the amino acid composition and properties of emerging and old sub lineage GIV isolates in the C protein region are significantly changed (Appendix A). The functional study of the C protein found that the hydrophobic domain and high proportion of basic amino acids of the C protein are crucial in the formation of nucleocapsid.

The JEV E protein is a membrane fusion protein, which is responsible for virus adsorption, penetration, and pathogenicity. According to previous studies, mutations in Domain III, 40 and 41 have significant effect for changing the affinity to the antibody [6,7]. Among mutations in the E protein of JEV GIV, ED3 40 and 41 retained no mutation with standard. Amino acid E327 in the emerging and old sub lineage isolates have an amino acid mutation from hydrophobic leucine to hydrophilic glutamine. This site is located on the surface of the side of Domain III and is associated with receptor binding. The change of the emerging and old sub lineage isolates at this site may affect the recognition of the host lipid membrane by the E protein and the membrane binding process, thus changing the emerging sub lineage GIV JEVs’ host or vector preference. This makes the emerging sub lineage GIV JEVs more adaptable to the new host and environment. Nevertheless, further experiments are needed to confirm the effect of the mutations in these key sites on JEVs.

JE is a vaccine-preventable epidemic, and vaccine is the most effective means to deal with the epidemic of the Japanese encephalitis virus. However, the current JE vaccine, whether an attenuated live vaccine or inactivated vaccine, is prepared with JEV GIII [53]. Recent studies have shown that the current JE vaccine can effectively protect against the JE virus GIII, but cannot fully protect against JEV GI [54]. The protection efficiency of the current JE vaccine against JEV GV virus is even lower [55]. However, further experiments are needed to assess the protection efficiency of the current JE vaccine against JEV GIV. Furthermore, experimental studies between the emerging JEV GIV isolates and the current JE vaccine, including the challenge test for animal protection, are needed. In addition, the reemergence of genotype IV JEVs in Australia may indicate the possibility of future JE outbreaks, and thus it is necessary to enhance the detection and surveillance of the local JEV in mosquito vectors, host animals (pigs and birds), and humans.

## 5. Conclusions

These results demonstrate that JEV GIV is the youngest JEV genotype at a rapid evolution stage. Considering the emerging GIV JEVs have a good host/vector adaptability and transmissibility, surveillance of emerging GIV JEVs should be strengthened to prevent them from spreading to new areas and causing outbreaks of JEV GIV infection.

## Figures and Tables

**Figure 1 viruses-15-00626-f001:**
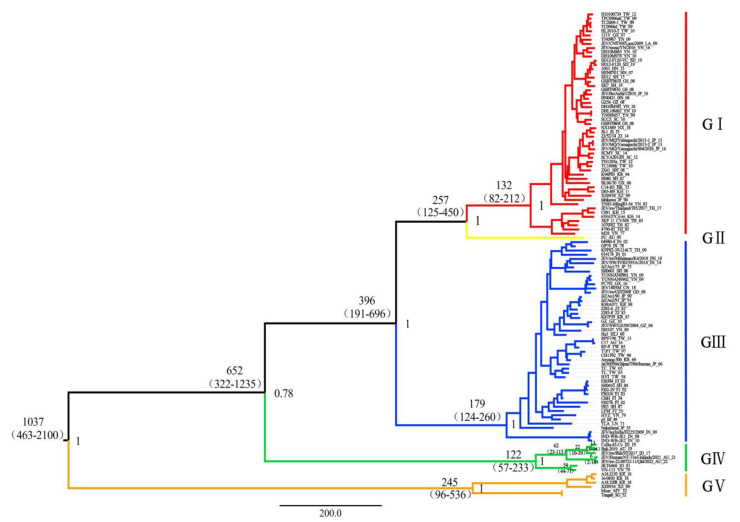
MCC tree for 115 whole genome sequences of the JEV. Five distinct sub lineages were identified: GI (red), GII (yellow), GIII (blue), GIV (green), and GV (orange). Estimated tMRCAs (time to most recent common ancestor) of these sub lineages (with their 95% HPD values in parentheses) are shown, GI:132 (82–212), GIII:179 (124–260), GIV:122 (57–233), and GV:245 (96–536). JEV GII contains only one virus and therefore does not estimate the TMRCA. The posterior probability value of each cluster was shown on the right of the node.

**Figure 2 viruses-15-00626-f002:**
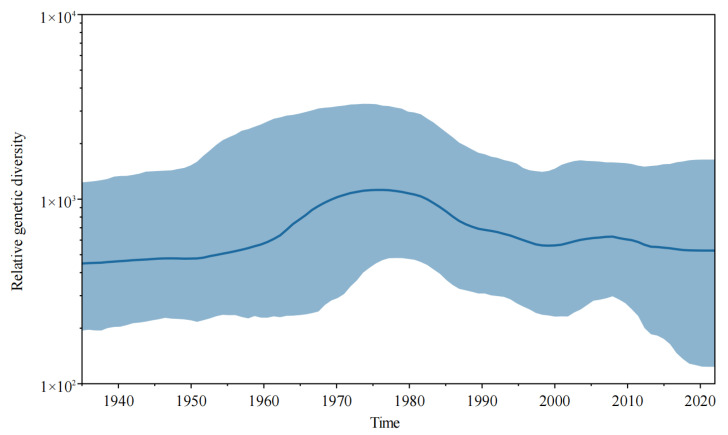
Bayesian skyline plot for the JEV. Highlighted areas correspond to 95% HPD intervals. The horizontal and vertical axes represent time and genetic diversity, respectively.

**Figure 3 viruses-15-00626-f003:**
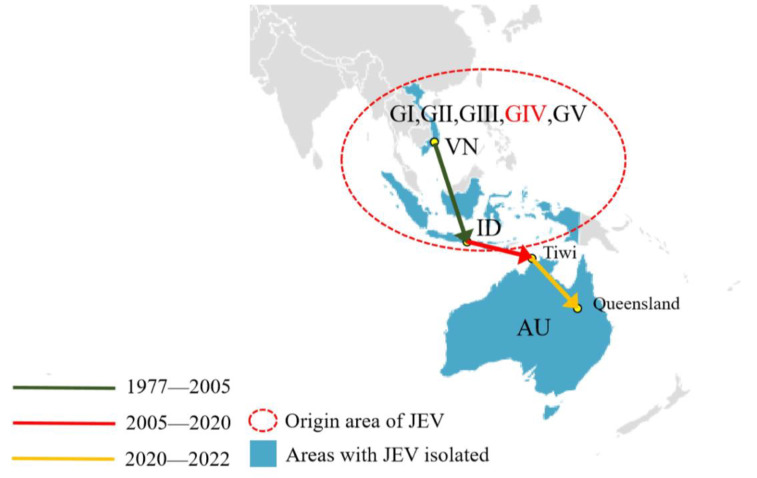
The spatiotemporal migration of JEV GIV since the 1970s. Blue shadows represent areas with JEV isolation. Lines and arrows indicate the migration route and direction of the JEV. Green line: spread from Vietnam to Indonesia from 1977 to 2005; red line: spread from Indonesia to Australia’s Tiwi Islands from 2005 to 2020; orange line: spread from the Australian Tiwi Islands to Queensland from 2020 to 2022. The red dotted area shows the origin of JEV, including five genotypes. VN: Vietnam; ID: Indonesia; AU: Australia; Tiwi: Australia: Northern Territory, Tiwi Islands; Queensland: Australia: Queensland.

**Figure 4 viruses-15-00626-f004:**
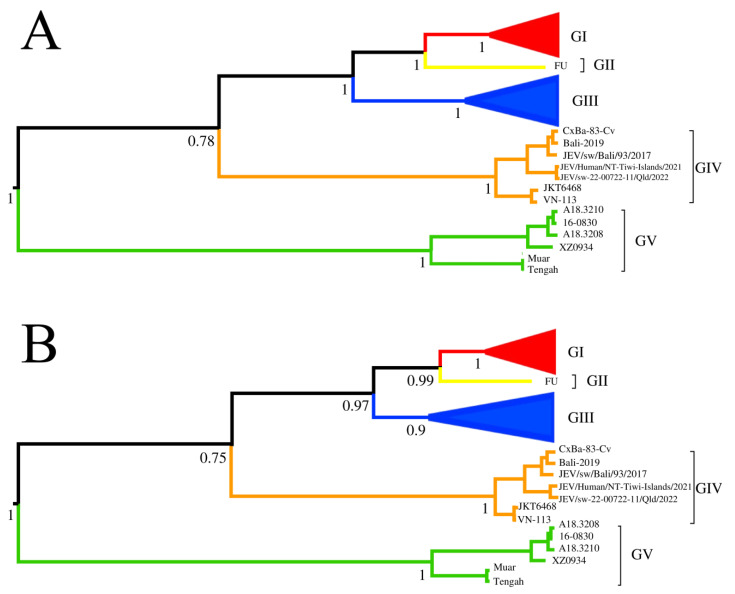
Phylogenetic analysis of the JEV. (**A**) The phylogenetic tree based on the whole genome sequences of JEVs. (**B**) The phylogenetic tree based on the sequences (1–300 nt) of domain I (VR) within the 3′ UTRs of the JEV. Red, yellow, blue, orange, and green were used to mark GI, II, III, IV, and V of JEVs, respectively. The triangles were used to condense strains of the same genotypes. The posterior probability value of each cluster was shown on the left of the node.

**Figure 5 viruses-15-00626-f005:**
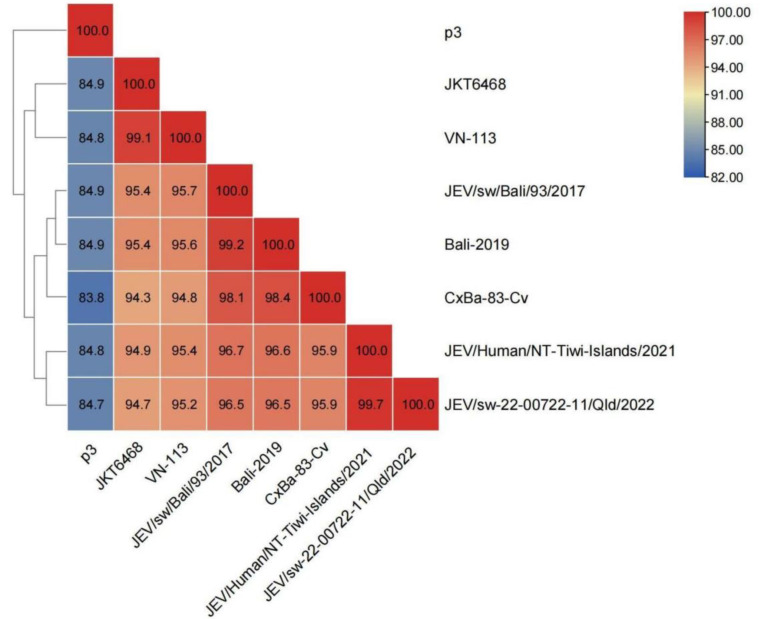
Heat map of the nucleotide similarity between JEV GIV isolates and vaccine strain P3. The value represents the nucleotide similarity between strains, and the similarity is shown from low to high in blue to red. The virus sequence correlation is represented by the left tree.

**Figure 6 viruses-15-00626-f006:**
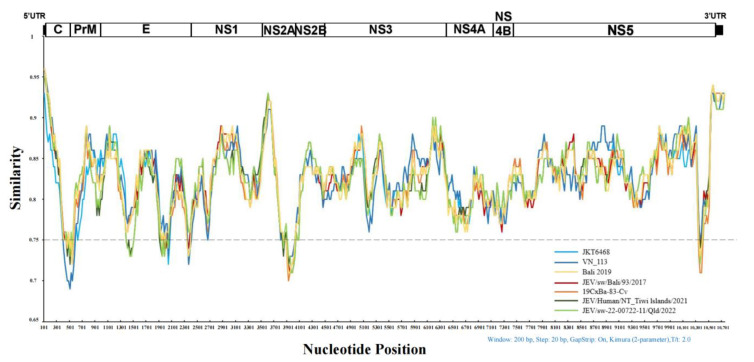
Plots of the nucleotide similarity between JEV GIV isolates with vaccine strain p3 as baseline. The upper limit of similarity is 1, and the lowest value is 0.65. Each curve is a comparison between the JEV GIV and p3 as reference. The two different blue colors represent the old sub lineage of JEV GIV, the line colored by yellow to red represent the emerged strains isolated from Bali Island until 2019, and the two different green colors represent the emerged strain isolated in Australia in 2021 and 2022.

**Figure 7 viruses-15-00626-f007:**
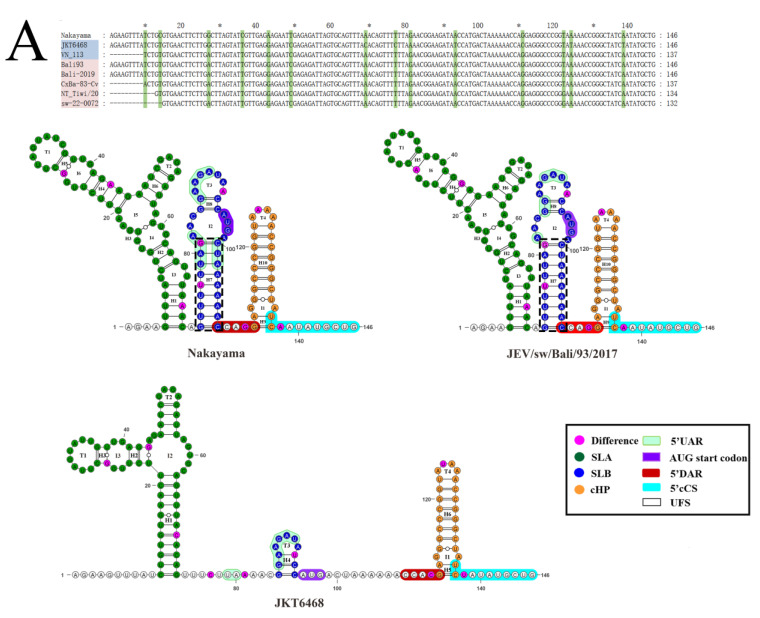
The primary sequence and secondary structure of the 5′UTR (**A**) and 3′UTR (**B**) of the JEV. In the primary sequence, the names of the strains are marked with blue and red shades, representing the old sub lineage and the emerging sub lineage of JEV GIV isolates. The mutations within the JEV isolates were labeled in green. The * is a replacement for odd mark number on top of sequences. Abbreviated names have been used to represent four isolates: Bali93, JEV/sw/Bali/93/2017; Cx-Ba-83-Cv,19CxBa-83-Cv; NT_Tiwi/20, JEV/Human/NT_Tiwi-Islands/2021; and sw-22-0072, JEV/sw-22-00722-11/Qld/2022. In the secondary structure, three strains of JEV isolates (Nakayama, JEV/sw/Bali/93/2017 and JKT6468), which represent the JEV reference strain and the representative strains for the emerging and old sub lineages of JEV GIV, respectively. The color and annotation labeled in the secondary structures represent different structural elements or cis-acting elements. In (**B**), the length of 3 ‘UTRs varies greatly among strains, but all can be divided into three main domains: Domain I, Domain II, and Domain III, which were indicated with light gray, dark gray, and black bars.

**Figure 8 viruses-15-00626-f008:**
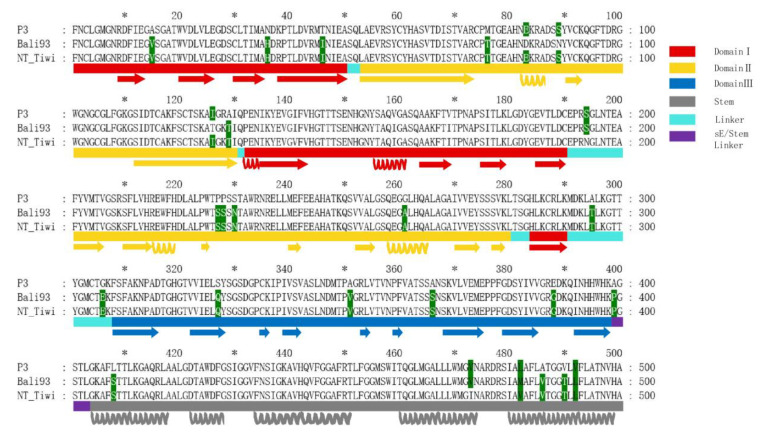
Envelope protein structural and sequence alignment of the JEV GIV emerging strains and vaccine strain P3. The * is a replacement for odd mark number on top of sequences. The alignment of the E protein sequences is above shapes, and the highlighted amino acid mutation sites are shown. The shapes under the sequences are the corresponding protein structure information. The length and position of the secondary structure shapes correspond to the sequence alignment strictly, which could indicate their precise information in the model. The starting position of the lower secondary structure graph strictly corresponds to the predicted model, and the highlighted amino acid mutation sites are shown in the sequence.

**Figure 9 viruses-15-00626-f009:**
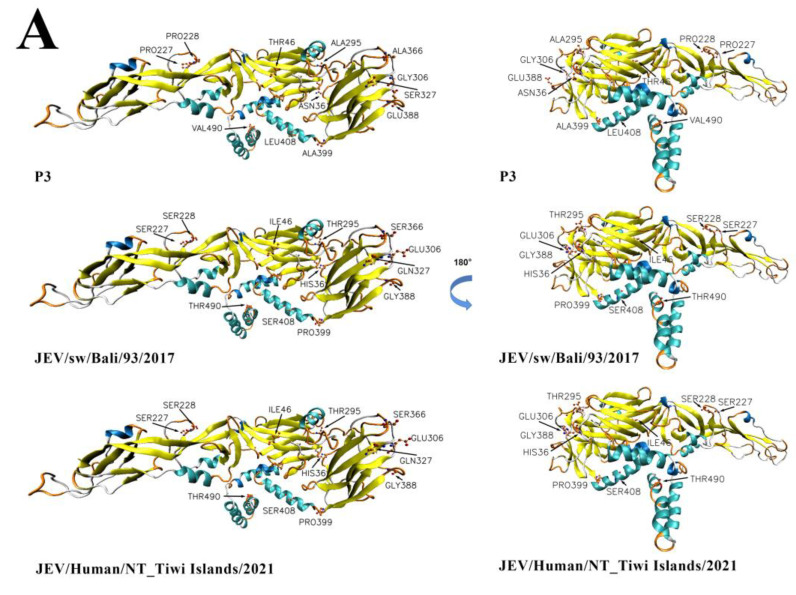
Model of the three-dimensional structure and electrostatic potential in JEV GIV and P3 strains. In the secondary structure, three strains of JEV isolates (Nakayama, JEV/sw/Bali/93/2017 and JKT6468) represent the JEV standard strain and the representative strains for the emerging and old sub lineages of JEV GIV, respectively. (**A**) The three-dimensional structure and mutation sites diagram. C: orange2 (1.000, 0.390, 0.000) N: blue (0.000, 0.000, 1.000) O: red (1.000, 0.000, 0.000) Alpha Helix:cyan (0.210, 0.750, 0.750) 3_ 10_ Helix:blue2 (0.020, 0.460,0.800) Pi_ Helix:red (1.000, 0.000, 0.000) Extended_ Beta:yellow (1.000, 1.000, 0.000) Bridge_ Beta:tan (0.500, 0.500, 0.200) Turn:orange (1.000, 0.500, 0.000) Coil:White (1.000, 1.000, 1.000) (**B**) Surface charge distribution map of the E protein structure. Rectangle on the left shows the crowded mutations in Domain III, including G306E, S327Q, A366S, and E388. The rectangle on the right shows the mutation, V490T, which is located at the linker of sE and helical stem and anchor regions.

**Table 1 viruses-15-00626-t001:** The JEV isolates analyzed in the current study.

NO	Strains	Date	Locations	Genotype	Locus
1	M28	1977	China	I	JF706279.1
2	1070/82 (Subin)	1982	Thailand	I	GQ902059.1
3	YN83-Meng83–54	1983	China: Yunnan	I	JF706282.1
4	3KP’’U’’CV569	1985	Thailand	I	GQ902060.1
5	4790-85	1985	Thailand	I	GQ902062.1
6	Ishikawa	1994	Japan	I	AB051292.1
7	K94P05	1994	Korea	I	AF045551.2
8	SH-80	2001	China: Shanghai	I	JN381848.1
9	HN0421	2004	China: Henan	I	JN381841.1
10	BL06-50	2006	China: Guangxi	I	JF706270.1
11	131V	2007	China: Guangxi	I	GU205163.1
12	HEN0701	2007	China: Henan	I	FJ495189.1
13	JX61	2008	China: Henan	I	GU556217.1
14	GSBY0816	2008	China: Gansu	I	JN381842.1
15	GSBY0804	2008	China: Gansu	I	JN381844.1
16	GSBY0810	2008	China: Gansu	I	JN381840.1
17	GZ56	2008	China: Guizhou	I	HM366552.1
18	YN09M57	2009	China: Yunnan	I	KT229574.1
19	JEV/CNS769/Laos/2009	2009	Laos	I	KC196115.1
20	YN0967	2009	China: Yunnan	I	JF706268.1
21	JEV/Taiwan/TC0906d/M/2009	2009	China: Taiwan	I	KF667320.1
22	JEV/Taiwan/TPC0906ah/M/2009	2009	China: Taiwan	I	KF667318.1
23	XZ0938	2009	China: Xizang	I	HQ652538.1
24	TC2009-1	2009	China: Taiwan	I	JF499790.1
25	DH10M978	2010	China: Yunnan	I	KT229573.1
26	DH10M865	2010	China: Yunnan	I	KT229572.1
27	DHL10M62	2010	China: Yunnan	I	KT229575.1
28	SCCZ	2010	China: Sichuan	I	KU351667.1
29	HL2010-2	2010	China: Taiwan	I	JQ031753.1
30	JEV/Bo/Aichi/1/2010	2010	Japan: Aichi	I	AB853904.1
31	JEV/Taiwan/TC1006h/M/2010	2010	China: Taiwan	I	KF667321.1
32	DH10M585	2010	China: Yunnan	I	KT957421.1
33	SCYA201201	2012	China: Sichuan	I	KM658163.1
34	JEV/Taiwan/TN1205a/M/2012(2)	2012	China: Taiwan	I	KF667325.1
35	JEV/Taiwan/H10100739/H/2012	2012	China: Taiwan	I	KF667324.1
36	JEV/MQ/Yamaguchi/2013/2	2013	Japan: Yamaguchi, Yoshida	I	AB981184.1
37	JEV/MQ/Yamaguchi/2013/1	2013	Japan: Yamaguchi, Yoshida	I	AB981183.1
38	10S3	2013	China: Henan	I	MF542268.1
39	SCMY	2014	China: Sichuan	I	KU351668.1
40	ZJ/52/14	2014	China	I	MK558811.1
41	639A37Cx-tri	2014	Cambodia	I	KY927815.1
42	JS-1	2015	China: Jiangsu	I	KX357114.1
43	C081	2015	Cambodia	I	KY927816.1
44	C14-B3	2015	Cambodia	I	KY927817.1
45	D03-B9	2015	Cambodia	I	KY927818.1
46	SD12	2015	China: Shanghai	I	MH753127.1
47	JEV/MQ/Yamaguchi/804/2016	2016	Japan: Yamaguchi, Yoshida	I	LC461957.1
48	JEV/mosq/YN/2016	2016	China	I	MH385014.1
49	SH7	2016	China: Shanghai	I	MH753129.1
50	JEV/sw/Thailand/185/2017	2017	Thailand	I	LC461958.1
51	NX1889	2018	China: Ningxia	I	MT134112.1
52	SD12-F120	2019	China	I	MN544779.1
53	SD12-F120-VC	2019	China	I	MN544780.1
54	FU	1995	Australia	II	AF217620.1
55	Nakayama	1935	Japan	III	EF571853.1
56	p3	1949	China: Beijing	III	U47032.1
57	CBH	1954	China: Fujian	III	JN381860.1
58	LFM	1955	China: Fujian	III	JN381863.1
59	HVI	1965	China: Taiwan	III	AF098735.1
60	Ha3	1960′s	China: Heilongjiang	III	JN381872.1
61	TL	1965	China: Taiwan	III	AF098737.1
62	TC	1965	China: Taiwan	III	AF098736.1
63	JaOH0566/Japan/1966/human	1966	Japan	III	AY508813.1
64	Anyang-300	1969	South Korea	III	KT447437.1
65	TLA	1971	China: Liaoning	III	JN381868.1
66	JaTAn1/75	1975	Japan: Tokyo	III	AB551990.1
67	GP78	1978	India	III	AF075723.1
68	HYZ	1979	China: Yunnan	III	JN381853.1
69	ZJ82-6	1982	China: Zhejiang	III	KY650724.1
70	ZJ83-8	1983	China: Zhejiang	III	KY650725.1
71	RP-9	1985	China: Taiwan	III	AF014161.1
72	SH3	1987	China: Shanghai	III	JN381864.1
73	K87P39	1987	South Korea	III	AY585242.1
74	K88A071	1988	South Korea	III	KR908703.1
75	DH107	1989	China: Yunnan	III	JN381873.1
76	CH1392	1990	China: Taiwan	III	AF254452.1
77	JaTAn1/90	1990	Japan: Tokyo	III	AB551991.1
78	JaTAn2/91	1991	Japan: Tokyo	III	AB551992.1
79	T1P1	1997	China: Taiwan: Liu-Chiu islet	III	AF254453.1
80	14178	2001	India	III	EF623987.1
81	Fj02-29	2002	China: Fujian	III	JF706273.1
82	04940-4	2002	India: Maharashtra, Bhandara district	III	EF623989.1
83	Fj0276	2002	China: Fujian	III	JN381867.1
84	FJ0339	2003	China: Fujian	III	JN381859.1
85	FJ0394	2003	China: Fujian	III	JN381858.1
86	SH0410	2004	China: Shanghai	III	JN381856.1
87	JEV/SW/GZ/09/2004	2004	China: Guizhou	III	KF297916.1
88	SH0601	2006	China: Shanghai	III	EF543861.1
89	IND-WB-JE1	2008	India: West Bengal	III	JX050179.1
90	JEV/sw/GD/2008	2008	China: Guangdong	III	KX965684.1
91	KPP82-39-214CT	2009	Thailand	III	GQ902063.1
92	JEV/eq/India/H225/2009	2009	India	III	JX131374.1
93	YUNNAN0902	2009	China: Yunnan Province	III	JQ086763.1
94	YUNNAN0901	2009	China: Yunnan	III	JQ086762.1
95	IND-WB-JE2	2010	India: West Bengal	III	JX072965.1
96	GZ	2010	China: Guizhou	III	KC915016.1
97	RP9-190	2013	China: Taiwan	III	KF907505.1
98	JEV/SW/IVRI/395A/2014	2014	India	III	KP164498.2
99	C17	2016	Angola	III	KX945367.1
100	FC792	2016	China: Guangxi	III	MF002373.1
101	JEV1805M	2018	China	III	MN639770.1
102	JEV/sw/Mindanao/K4/2018	2018	Philippines: Mindanao	III	LC461960.1
103	VN 113	1979	Viet Nam	IV	KU705228.1
104	JKT6468	1981	Indonesia	IV	AY184212.1
105	JEV/sw/Bali/93/2017	2017	Indonesia: Bali, Denpasar	IV	LC461961.1
106	Bali 2019	2019	Australia	IV	MT253731.1
107	19CxBa-83-Cv	2019	Indonesia: Bali	IV	LC579814.1
108	JEV/Human/NT_Tiwi Islands/2021	2021	Australia: Northern Territory, Tiwi Islands	IV	OM867669.1
109	JEV/sw-22-00722-11/Qld/2022	2022	Australia: Queensland	IV	ON624132.1
110	Muar	1952	Malaysia	V	HM596272.1
111	Tengah	1952	Singapore	V	KM677246.1
112	XZ0934	2009	China: Xizang	V	JF915894.1
113	A18.3210	2018	South Korea: Camp Humphreys	V	MT568538.1
114	A18.3208	2018	South Korea: Camp Humphreys	V	MT568539.1
115	16-0830	2016	South Korea: Yongsan	V	MT568540.1

## Data Availability

The original contributions presented in the study are included in the article/Appendix A, further inquiries can be directed to the corresponding authors.

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
