# Peer review of "Re-Emerged Genotype IV of Japanese Encephalitis Virus Is the Youngest Virus in Evolution"

_viruses, 2023, doi:10.3390/v15030626_

Round 1
Reviewer 1 Report
Review: viruses-2216348
This manuscript by Xu et al described the structure of JEV genotypes and in particular the molecular evolution of JEV GIV lineage. While the subject is appropriate for the journal, publication of this manuscript in its present form is not recommended, because it contains a number of unexplained observations.
To be considered further for publication the manuscript will need to be more organized (including adequate references) and elaborative in support of the claims made in the paper. Some specific points of concern are noted below:
1) There have, of course, been many other efforts to computationally map the structural consequences of mutations in the JEV. These have largely been overlooked in the current work. It is a general professional courtesy expected of authors to at least recognize the previously published primary works on the topic, especially if submitted work follows the framework of similar earlier studies. Specific studies of JEV mutational analysis should be cited. See for example:
https://doi.org/10.1007/s12026-020-09130-y
https://doi.org/10.3390/zoonoticdis2030012
2) In the abstract and discussion sections the authors claim the uniqueness of this JEV GIV lineage. Briefly explain it with more clarity in the discussion section.
3) The methods section seems scattered. It needs to be concise and structured.
4) If a mutation makes a more stable form of the JEV GIV does that make this mutant protein bind with more affinity to the antibody or will it bind with less affinity? A discussion would make this clearer to the reader ahead of the conclusion and should be in the discussion.
3) Figure 1, 4 and 7B should be clearer
4) A zoomed in structural view for JEV GIV Domain III is suggested. The color code for Figure 9 is necessary.
Reviewer 2 Report
Xu et al., elaborated the story of JEV GIV isolations in Australia from a recent epidemic, sequenced the isolates, studied them by various methods, phylogenetic trees etc. The work helps to understand better the JE situation in Australia, makes the basis for further local experimental studies and epidemiological measures. The English language, the applied methods, their interpretations, references are correct.
Line numbers are missing, it is more difficult to indicate the exact place of changes/mistakes that way.
Introduction.
Page 1, last paragraph – Flaviviridae subfamily, Flavivirus genus
Page 2. 1. paragraph, ref [6] ticks are vectors or host ????
- Korea (South or North?)
- Space after [17]
- what is VE ? Has not been given yet in full.
Mat and methods, 2.2. included in what? in the study/analyses etc.
Results. paragraph 1 what is tMRCA? were given in written form?
- 3.6. and 3.7.1. 3.7.2., 3.7.3 These data should have been shown in separate small Tables. It is very difficult to follow and understand these data piles of data.
- Figures 6, 7, 8, 9 Figure legends should be much shorter, not a whole paragraph
Discussion.
2 paragraph. Tick-borne encephlitis virus is also an important zoonotic arbovrus of the Eurasian continent. About its evolutionary origin Kovalev and Mukchaceva wrote paper 2014 Ecol. evol.
3. paragraph. See similar articles about TBEV virus mutation rate Uzcategui et al., 2012, Egyed et al., 2018).
The W.-European subtype of TBEV appeared about 350 years ago (Kovalev and Mukhacheva 2012).
4. paragraph. The entire or most part of this paragraph should be part of Introduction.
Last paragraph. JE is not an epidemic but a clinical illness.
- Further experiments are needed to test whether present JEV vaccines protect against emerging novel JEV GIV strains.
- last sentence „ indicate the possibility of future JE outbrakes in Australia, ……
Questions:
What do You think happened 2-300 years ago, when Flavivirus evolution erupted almost at the same time at various parts of our planet?
How the virus could arrived to Austria? is there any explanation?
Spelling: Res. 3. 1 paragraph, last sentence dot, space.
2. paragraph – from porcine (XY), mosquito (CV), and human ( HJ) specimen. next sentence: „a patient or patients”
discussion 5 paragraph. A previous study…….
